# Hsp70 chaperones are non-equilibrium machines that achieve ultra-affinity by energy consumption

Paolo De Los Rios*, Alessandro Barducci*

Laboratoire de Biophysique Statistique, Ecole Polytechnique Fédérale de Lausanne (EPFL), Lausanne, Switzerland

**Abstract** 70-kDa Heat shock proteins are ATP-driven molecular chaperones that perform a myriad of essential cellular tasks. Although structural and biochemical studies have shed some light on their functional mechanism, the fundamental issue of the role of energy consumption, due to ATP-hydrolysis, has remained unaddressed. Here we establish a clear connection between the non-equilibrium nature of Hsp70, due to ATP hydrolysis, and the determining feature of its function, namely its high affinity for its substrates. Energy consumption can indeed decrease the dissociation constant of the chaperone-substrate complex by several orders of magnitude with respect to an equilibrium scenario. We find that the biochemical requirements for observing such *ultra-affinity* coincide with the physiological conditions in the cell. Our results rationalize several experimental observations and pave the way for further analysis of non-equilibrium effects underlying chaperone functions.

## Introduction

ATP-driven molecular chaperones play a central role in protecting cells against proteins that could unfold or misfold because of mutations, various stresses or fluctuations and ultimately result in cyto-toxic aggregates (*Bukau et al., 2006*; *Hartl et al., 2011*). 70-kDa Heat shock proteins (Hsp70s) stand out for several reasons: they are possibly the most ubiquitous, they function as monomers, and they supervise a plethora of diverse cellular processes (*Mayer and Bukau, 2005*; *Zuiderweg et al., 2013*) such as protein translation (*Kramer et al., 2009*), protein trafficking (*Matlack et al., 1999*; *Neupert and Brunner, 2002*), the disassembly of protein complexes (*Böcking et al., 2011*), signaling (*Pratt and Toft, 2003*) and protein degradation (*Hohfeld et al., 2001*). All these tasks crucially depend on the high-affinity binding of Hsp70s to substrate proteins during a complex ATP-driven conformational cycle (*Mayer, 2013*). The ATP- and ADP-bound states of Hsp70 (Hsp70·ATP and Hsp70·ADP respectively) and their interconversion play a major role in the chaperone functional cycle: the nature of the bound nucleotide affects the affinity of the chaperone for its substrates, with Hsp70·ADP binding the substrate more stably than Hsp70·ATP (*Schmid et al., 1994*; *Theyssen et al., 1996*; *Packschies et al., 1997*; *Gisler et al., 1998*; *Russell et al., 1998*; *Laufen et al., 1999*; *Mayer et al., 2000*).

Intriguingly, several experimental evidences suggested that the effective affinity of Hsp70 for substrates when the chaperone was running through its ATP-hydrolysis driven cycle was significantly higher than both the ones of Hsp70·ATP and Hsp70·ADP (*Laufen et al., 1999*; *Wittung-Stafshede et al., 2003*). Because this remarkable result would not be possible within the boundaries of thermodynamic equilibrium, it is therefore necessary to clarify how ATP hydrolysis, and thus energy consumption, can affect the binding strength of Hsp70s to their substrates.

## Results and discussion

According to the consensus Hsp70 cycle (*Figure 1*) substrate binding/unbinding takes place with rates that depend on the state of the bound nucleotide ($k_{ATP}^{on}$, $k_{ATP}^{off}$, $k_{ADP}^{on}$, $k_{ADP}^{off}$) (*Schmid et al., 1994*; *Gisler*

*For correspondence: paolo. delosrios@epfl.ch (PD); alessandro.barducci@epfl.ch (AB)

**Competing interests:** The authors declare that no competing interests exist.

**Reviewing editor**: Arup K Chakraborty, Massachusetts Institute of Technology, United States

**eLife digest** Proteins perform numerous essential tasks in cells. Most of these tasks require the protein to have a very specific structure, which is maintained by a balance of chemical and physical interactions. However, this delicate balance is vulnerable to excessive heat, changes in the pH of the cell, and certain chemicals. As a consequence, proteins could lose their specific structure and stop working.

Cells employ a group of specialized proteins—called chaperones—to check that other proteins have the correct structure, and to 'refold' those that do not. The Hsp70 chaperone family needs energy to do its job, and it gets this energy from a molecule called ATP. However, the exact way that Hsp70s work and use this energy is not fully understood.

One major puzzle is how Hsp70 binds to a protein to fold it up. Previous experiments suggested that this binding is particularly effective if Hsp70 can adopt different structures as part of a complex cycle governed by ATP. Now, De Los Rios and Barducci reveal that the energy released from breaking down ATP molecules enables this extra-efficient binding to occur. According to the proposed model, this is possible under some conditions that are normally found in cells. These include having many more Hsp70 proteins than target proteins, and producing energy at extremely high rates from ATP. The specific kinetic properties of the different structures Hsp70 can form are also crucial.

More generally, the principle that energy consumption enhances binding could be extended beyond chaperone proteins and represent a general mechanism for other biomolecular systems.

*et al., 1998*; *Mayer et al., 2000*), with Hsp70·ATP exchanging the substrate two to three orders of magnitude faster than Hsp70·ADP. The conversion from Hsp70·ADP to Hsp70·ATP occurs through a nucleotide exchange process which is described here at an effective level as a simple first-order reaction with rate $k_{DT}$, or $k_{DT}^S$ in the presence of a bound substrate (*Figure 1*, and 'Materials and methods' for a full derivation). The conversion from Hsp70·ATP to Hsp70·ADP can occur by means of two different processes: nucleotide exchange (dashed arrows in *Figure 1*, with rate $k_{TD}^{ex}$, and $k_{TD}^{ex,S}$ in the presence of the substrate) and ATP hydrolysis (red arrows in *Figure 1*), whose rate depends on the absence or presence of a bound substrate ($k_h$ and $k_h^S$ respectively). The total conversion rate from Hsp70·ATP to Hsp70·ADP is thus $k_{TD} = k_{TD}^{ex} + k_h$ (and analogous expressions in the presence of a substrate). In the cell, several cochaperones tune the exchange and hydrolysis rates: J-domain proteins (JDPs) enhance the rate of ATP hydrolysis, and nucleotide exchange factors (NEFs) catalyze nucleotide release (*Youker and Brodsky, 2007*; *Kampinga and Craig, 2010*). Within the present description, cochaperones are not taken into account explicitly. Rather, their action is captured as a modulation of the cycle timescales. In particular, JDPs are known to bind the substrate and subsequently interact with Hsp70, enhancing ATP hydrolysis. Consequently here only the hydrolysis rate in the presence of the substrate, $k_h^S$, is affected by the action of JDPs.

A significant difference between hydrolysis and nucleotide exchange must be outlined here: ATP-hydrolysis, at variance with exchange, results into a net production of ADP and a loss of ATP. In a cellular perspective, the ATP and ADP concentrations are kept fixed by energy-consuming chemostats. In vivo, ATP hydrolysis is therefore an intrinsically non-equilibrium process.

Because we aim here at elucidating the relation between energy consumption and substrate affinity, we determine the effective dissociation constant of the system, $K_{eff}$, which provides a coarse-grained measure of how well Hsp70s can bind their substrates through their cycle. $K_{eff}$ is defined in the usual way as $K_{eff} = [S][Hsp70]/[Hsp70·S]$, where $[Hsp70]$ is the total concentration of Hsp70 not bound to a substrate ($[Hsp70] = [Hsp70·ATP]+[Hsp70·ADP]$), $[Hsp70·S]$ is the total concentration of substrate-bound chaperone ($[Hsp70·S] = [Hsp70·ATP·S]+[Hsp70·ADP·S]$) and $[S]$ is the concentration of free substrate.

In the absence of hydrolysis, no energy is consumed and all the reactions of the Hsp70 cycle are at equilibrium. In this scenario, where all the reactions are driven by thermal fluctuations, the *detailed balance* rule holds and each branch of the biochemical cycle is individually balanced ('Materials and methods'). In fact, in this case, the ratio between the forward and backward rates for each reaction

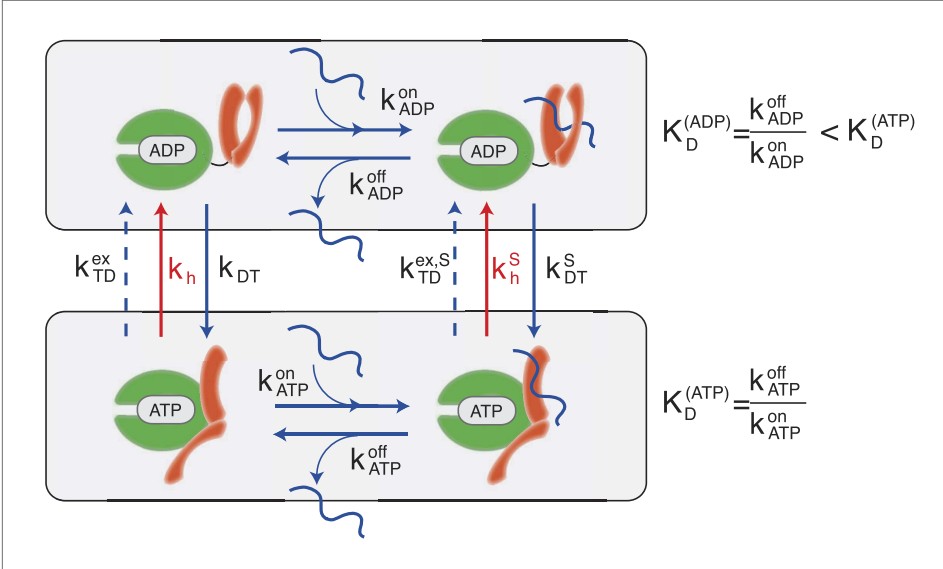

**Figure 1**. Canonical Hsp70 biochemical cycle. The model takes into account four states in Hsp70 (NBD is schematically represented here in green, SBD in orange), which are defined by substrate binding and by the nature of the bound nucleotide (ADP or ATP). The rates of the substrate binding/unbinding process (horizontal blue lines) are influenced by the nucleotide ($k_{ATP}^{on}$, $k_{ATP}^{off}$, $k_{ATP}^{on}$, $k_{ATP}^{off}$). ADP-bound states are converted to ATP-bound states through a nucleotide exchange process (vertical solid blue lines) with rates $k_{DT}$, $k_{DT}^{S}$. The ATP to ADP conversion can occur by means of either a nucleotide exchange process (dashed blue lines) with rates $k_{TD}^{ex}$, $k_{TD}^{ex,S}$ or ATP-hydrolysis (red lines) with rates $k_h$, $k_h^S$.

is completely determined by the free energy difference between the two states, for example $k_{DT}^{S}/k_{TD}^{S} = \exp\left(G_{\text{Hsp70·ADP·S}} - G_{\text{Hsp70·ATP·S}}\right)$. The equilibrium effective dissociation constant ($K_{\text{eff}}^{eq}$) can be easily determined as

$$K_{\text{eff}}^{eq} = \frac{k_{DT}^{S}K_{D}^{(ATP)} + k_{TD}^{S}K_{D}^{(ADP)}}{k_{DT}^{S} + k_{TD}^{S}}$$

where $K_{D}^{(ATP)}$ and $K_{D}^{(ADP)}$ are the dissociation constants of the Hsp70·ATP-substrate and Hsp70·ADP-substrate complexes, respectively. Not surprisingly, $K_{\text{eff}}^{eq}$ corresponds to a weighted average of $K_{D}^{(ATP)}$ and $K_{D}^{(ADP)}$, and it cannot be lower than $K_{D}^{(ADP)}$, usually the lowest of the two (**Schmid et al., 1994**; **Gisler et al., 1998**; **Mayer et al., 2000**). As a matter of fact, in vivo $K_{\text{eff}}^{eq}$ would be close to its typical upper bound, namely $K_{D}^{(ATP)}$, because of the cellular excess of ATP over ADP. The equilibrium dissociation constant can be in principle measured in experiments where Hsp70 hydrolysis deficient mutants are used (**McCarty and Walker, 1991**), at varying ratios of the concentrations of the two nucleotides.

At variance with equilibrium, when ATP hydrolysis is turned on, the energy budget along the cycle is not restricted to the free-energy differences between the different states. Rather, the dissipated energy $E_{\text{diss}}$, which is related to ATP hydrolysis, must be taken into account. Thus *detailed balance* is broken and pairwise reactions cannot be solved individually as in the equilibrium case.

Nonetheless, even in non-equilibrium conditions a steady-state solution of the cycle exists ('Materials and methods'), and it provides an expression for the non-equilibrium dissociation constant ($K_{\text{eff}}^{neq}$).

$$K_{\text{eff}}^{neq} = \frac{\left(k_{DT}^{S}k_{ATP}^{off} + k_{TD}^{S}k_{ADP}^{off} + k_{ATP}^{off}k_{ADP}^{off}\right)\left(k_{DT} + k_{TD}\right) + \left(k_{DT}^{S}k_{ATP}^{off}k_{ADP}^{on} + k_{TD}^{S}k_{ADP}^{off}k_{ATP}^{on}\right)[S]}{k_{DT}\left(k_{ADP}^{off} + k_{DT}^{S} + k_{TD}^{S}\right)k_{ATP}^{on} + k_{TD}\left(k_{ATP}^{off} + k_{DT}^{S} + k_{TD}^{S}\right)k_{ADP}^{on} + \left(k_{DT}^{S} + k_{TD}^{S}\right)k_{ATP}^{on}k_{ADP}^{on}[S]}$$

In this scenario, ATP hydrolysis is controlled by the fixed basal hydrolsysis rate $k_h$ and the substrate-enhanced rate $k_h^S$, which is further modulated in cellular conditions by JDP co-chaperones. The ratio $k_h^S/k_h$, that measures the hydrolysis acceleration, is thus a natural parameter to characterize the behavior of the system.

In order to prove the intimate relation between this quantity and the total energy consumption, we report in *Figure 2A* the experimentally-measurable hydrolysis flux, $P_{diss}$, defined as

$$P_{diss} = k_h \left[Hsp70 \cdot \text{ATP}\right] + k_h^S \left[Hsp70 \cdot \text{ATP} \cdot S\right]$$

as a function of $k_h^S/k_h$. Here we consider the experimentally determined parameters for the *Escherichia coli* DnaK-DnaJ system (see *Table 1*), and concentrations that roughly mimic cellular conditions ([Hsp70]$_{tot}$ = 40 μM and substoichiometric substrate, here [S]$_{tot}$ = 4 μM). Not surprisingly, the hydrolysis flux increases with $k_h^S/k_h$, before saturation, corresponding to a regime dominated by the rate-limiting exchange process (*Gassler et al., 2001*).

In *Figure 2B* we report $K_{eff}^{neq}$, as a function of the acceleration ratio $k_h^S/k_h$. As $k_h^S/k_h$ increases and more energy is consumed, $K_{eff}^{neq}$ decreases, until it becomes lower than $K_D^{(ADP)}$ by several orders of magnitude. Non-equilibrium conditions lead thus to a dramatic increase of the affinity of Hsp70s for their substrates, that could not be achieved at equilibrium, where the effective dissociation constant would be bounded between the ones of the ATP-bound and ADP-bound states. We dub such effect *ultra-affinity* in analogy with energy-consuming ultrasensitivity observed in many enzymatic systems (*Goldbeter and Koshland, 1981*).

The lower bound of the non-equilibrium dissociation constant is $K_{eff}^{neq} = k_{ADP}^{off}/k_{ATP}^{on}$, which can be achieved for extremely high values of $k_h^S/k_h$ (*Figure 2B*). This regime corresponds to the limiting case of substrate binding exclusively to Hsp70·ATP, which has the fastest binding rate, and being released exclusively from the ADP-bound state, which has the slowest unbinding rate. Our analysis indicates that this theoretical limit, recently hinted at (*Zuiderweg et al., 2013*), likely pertains to a regime that is not accessible to Hsp70s.

It must be stressed here that this lower bound, as well as ultra-affinity, depends on the kinetic properties of the cycle and does not rely on the dissociation constants of any nucleotide-bound state. In order to better elucidate this point, we move beyond the experimentally measured rates for the DnaK/DnaJ system, and we explore the theoretical dependence of $K_{eff}^{neq}$ on the time-scale separation between the binding/unbinding kinetics in the two states. To this aim, in *Figure 3A* we report $K_{eff}^{neq}$ as a function of both $k_h^S/k_h$ and the ratio $k_{ADP}^{off}/k_{ATP}^{off}$, which measures the time-scale separation in the unbinding kinetics, while keeping $K_D^{(ADP)}$ unchanged. For $k_{ADP}^{off} \geq k_{ATP}^{off}$ the non-equilibrium dissociation constant is bound to the equilibrium range. Ultra-affinity can be achieved only for $k_{ADP}^{off}/k_{ATP}^{off}<1$, and is more pronounced for larger time-scale separations. In the limit $k_{ADP}^{off} \ll k_{ATP}^{off}$, both the binding and the unbinding processes of the ADP-state become negligible and the non-equilibrium dissociation constant reduces to

$$K_{eff,0}^{neq} = K_D^{(ATP)} \frac{k_{DT}^S \left(k_{DT} + k_{TD}^{ex} + k_h\right)}{k_{DT} \left(k_{DT}^S + k_{TD}^{ex,S} + k_h^S\right)}$$

which is reported in *Figure 2B* (dashed line) and provides a good approximation of the exact behavior in the physiologically-accessible range of $k_h^S/k_h$.

Strikingly, ultra-affinity depends also on the concentration of the substrate as can be inferred from the explicit expression for $K_{eff}^{neq}$. This is another effect intrinsically tied to the non-equilibrium nature of the cycle. Indeed, at equilibrium, dissociation constants do not depend on the total species concentration since they simply encode the difference of free-energy between the bound and unbound states. We thus explore in *Figure 3B* the dependence of $K_{eff}^{neq}$ on both $k_h^S/k_h$ and the ratio between the total substrate and the total chaperone concentrations (for [Hsp70]$_{tot}$ = 40 μM). Two distinct regimes can be observed here with a sharp transition occuring at [S]$_{tot}$/[Hsp70]$_{tot}$ = 1. In the excess of chaperone ([S]$_{tot}$/[Hsp70]$_{tot}$<1) we observe ultra-affinity and the previously described behavior, whereas in the excess of substrate ([S]$_{tot}$/[Hsp70]$_{tot}$>1) the gain in affinity as a function of $k_h^S/k_h$ is limited and $K_{eff}^{neq}$ never exceeds its equilibrium range. This effect can be easily rationalized considering that in the latter condition, substrate binding to ADP-bound state becomes dominant and the system cannot exploit the time-scale separation in binding/unbinding kinetics to achieve ultra-affinity.

Quite surprisingly, the experimental characterization of the Hsp70-substrate dissociation constant in physiological, non-equilibrium conditions is extremely limited. However, ultra-affinity was implicitly suggested in a series of works assessing the binding of different substrates to Hsp70s, always in the presence of a co-localized JDP, thus ensuring maximal hydrolysis acceleration upon substrate binding (*Misselwitz et al., 1998*; *Laufen et al., 1999*; *Sullivan et al., 2001*). In all these assays the substrate

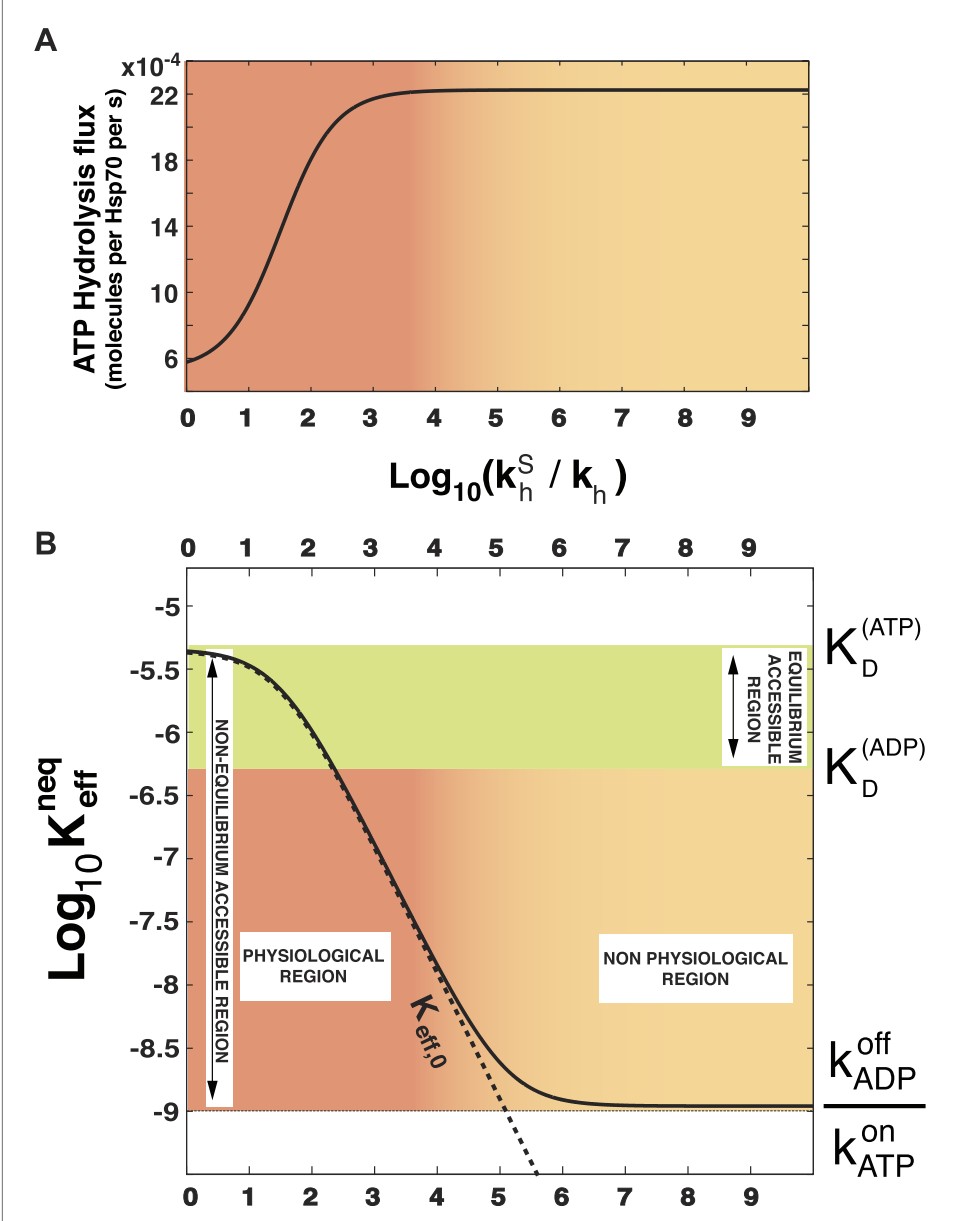

**Figure 2**. Effect of ATP-hydrolysis on $K_{eff}^{neq}$. Total energy consumption (**A**) and effective non-equilibrium dissociation constant of the Hsp70-substrate complex (**B**) is plotted as a function of the hydrolysis acceleration ratio $k_h^s/k_h$, for the DnaK/DnaJ/substrate system with concentrations $[Hsp70]_{tot}$ = 40 µM and $[S]_{tot}$ = 4 µM (see 'Materials and methods' for the parameters), The approximate dissociation constant $K_{eff,0}^{neq}$ is also plotted for comparison (black dashed line). The green region comprised between $K_D^{(ATP)}$ and $K_D^{(ADP)}$ corresponds to the range of affinities accessible at equilibrium (no hydrolysis). The red-to-yellow region corresponds to the values of the dissociation constants that are exclusively accessible to the non-equilibrium regime. The region where red fades to yellow ($10^3 \leq k_h^s/k_h \leq 10^4$) corresponds to the transition from physiological to non-physiological values of hydrolysis acceleration.

was observed to bind more efficiently in the presence of ATP, that drives the chaperone through its cycle, than in the presence of ADP, that instead blocks the system in the Hsp70·ADP state, the one with the lowest thermodynamic dissociation constant. We know of only one case where ultra-affinity has been carefully measured for the Hsp70 system interacting with a substrate peptide fused to a J-domain (***Wittung-Stafshede et al., 2003***). The dissociation constant observed with this setup (≈0.22 nM) was two orders of magnitude smaller than both the measured $K_D^{(ATP)}$ and $K_D^{(ADP)}$ (≥30 nM). If we combine our model with the specific rates provided in (***Wittung-Stafshede et al., 2003***) and values of $k_h^S/k_h$

**Table 1.** Parameters of the model

| | |
|---|---|
| $k_h$ | 0.0006 s$^{-1}$ (*McCarty et al., 1995*) |
| $k_{h.max}^s$ * | 1.8 s$^{-1}$ (*Laufen et al., 1999*) |
| $k_{ATP}^{on}$ | $4.5 \times 10^5$ s$^{-1}$ M$^{-1}$ (*Schmid et al., 1994; Gisler et al., 1998*) |
| $k_{ATP}^{off}$ | 2 s$^{-1}$ (*Schmid et al., 1994; Gisler et al., 1998*) |
| $k_{ADP}^{on}$ | 1000 s$^{-1}$ M$^{-1}$ (*Mayer et al., 2000*) |
| $k_{ADP}^{off}$ | $4.7 \times 10^{-4}$ s$^{-1}$ (*Mayer et al., 2000*) |
| $k_{ATP}^-$ | $1.33 \times 10^{-4}$ s$^{-1}$ (*Russell et al., 1998*) |
| $k_{ATP}^+$ | $1.3 \times 10^5$ s$^{-1}$ M$^{-1}$ (*Russell et al., 1998*) |
| $k_{ADP}^-$ | 0.022 s$^{-1}$ (*Theyssen et al., 1996; Russell et al., 1998*) |
| $k_{ADP}^+$ | $2.67 \times 10^5$ s$^{-1}$ M$^{-1}$ (*Russell et al., 1998*) |

Parameters used in the model, from various sources.
Notable cases are:
*This corresponds to the reaction Hsp70·ATP·DnaJ$_2$·S → Hsp70·ADP·DnaJ$_2$·S.

compatible to what reported in the literature, we obtain a predicted $K_{eff}^{neq}$ in the range 0.15–0.6 nM, which is in excellent agreement with the experimental value. All these findings strongly suggest that ultra-affinity becomes manifest when substrate binding is coupled with enhanced hydrolysis acceleration by the colocalization of the Hsp70 binding region and of a J-domain.

Our analysis of the cycle has unveiled the conditions that Hsp70 must satisfy to exhibit ultra-affinity: (*i*) the substrate-exchange rates of the ADP-bound state must be significantly slower than the ATP-state so that extremely different timescales can be exploited; (*ii*) ATP hydrolysis must be enhanced by orders of magnitude to fully enter the ultra-affinity regime; (*iii*) the chaperone must be in excess over the substrate. All these conditions are typically met in the cell by canonical Hsp70 chaperones, such as bacterial DnaK and cytosolic Hsc70 in eukaryotes: Hsp70·ADP is known to have limited exchange kinetics; Hsp70s are known to work only in partnership with JDPs; Hsp70s are highly abundant and typically in excess over JDPs (*Finka and Goloubinoff, 2013*). Cellular conditions seem thus to be optimal for ultra-affinity.

Our results provide an additional example of how ATP hydrolysis can be exploited by cells to overcome the constraints set by equilibrium thermodynamics. The key role of energy consumption in driving biochemical cycles for performing a variety of functions is well established (*Schnakenberg 1976; Hill, 2005; Ge et al., 2012*) and recently it has been recognized in cellular processes such as sensing, signaling and adaptation (*Qian and Reluga, 2005; Lan et al., 2012; Mehta and Schwab, 2012*).

Notably, the ultra-affinity concept proposed here shares some similarities with the well-known kinetic proofreading (*Hopfield, 1974*) since in both cases chemical energy consumption is used to

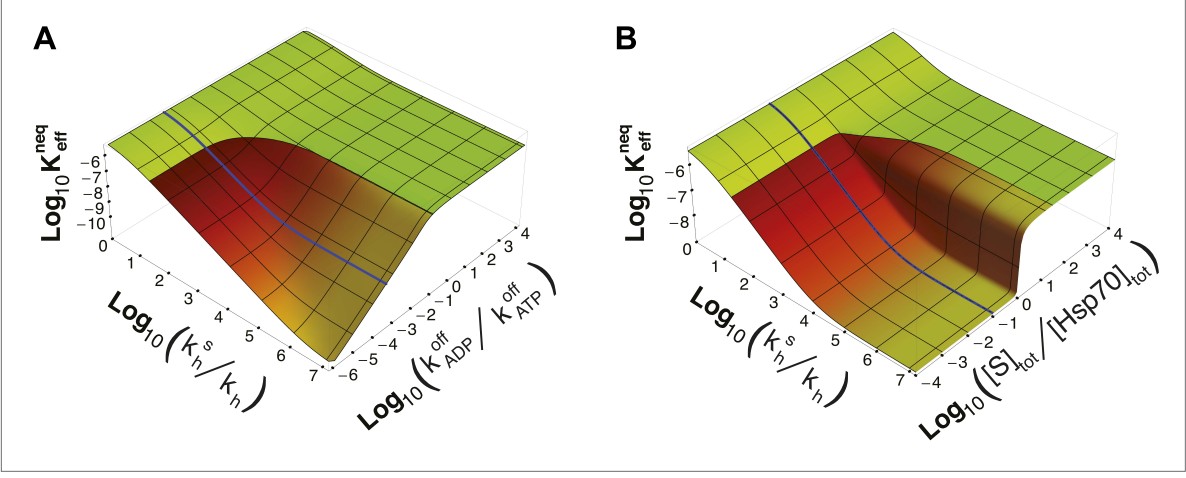

**Figure 3**. Dependence of $K_{eff}^{neq}$ on time-scale separation and on stoichiometric ratio. (**A**) Non-equilibrium dissociation constant as a function of the hydrolysis acceleration ratio $k_h^s/k_h$ and of the time-scale separation between the ATP- and ADP-state, expressed as the ratio between the substrate unbinding rates between the ATP- and ADP-state. (**B**) Non-equilibrium dissociation constant as a function of the hydrolysis acceleration ratio $k_h^s/k_h$ and of the stoichiometric ratio between the total substrate and Hsp70 concentrations. The color codes are the same as in **Figure 2**, green for the region accessible in equilibrium, and red-to-yellow for the region accessible in non-equilibrium. The blue line is the non-equilibrium dissociation constant reported in **Figure 2B**.

increase the binding affinity of specific molecules beyond their equilibrium value. However, a significant dissimilarity can be outlined: kinetic proofreading exploits multiple non-equilibrium steps to enhance a pre-existing difference in binding affinity among various substrates, whereas ultra-affinity is achieved due to the specific ability of the substrate to induce upon binding a non-equilibrium transition in the receptor (*i.e.*, the chaperone).

We expect that a similar approach may be applicable to more specialized Hsp70s, such as HscA (*Hesterkamp and Bukau, 1998*), and to other chaperones, with possibly more complex cycles, such as the GroEL/S system, Hsp100s and Hsp90s. As well, it would not be surprising to discover other molecular machines working along the same principles, so that the present study might also provide a general scheme applicable beyond Hsp70 chaperones.

## Materials and methods

We frame here the model in terms of deterministic mass-balance equations, which describe the evolution of the concentrations of the different states. This treatment is appropriate for the Hsp70 system since both chaperones and substrates are highly abundant in standard cellular conditions. In the case of less abundant molecular species fluctuations might not be negligible. Therefore a more detailed, stochastic description of the process would be required (*Ge et al., 2012*).

### Coarse graining of the nucleotide exchange processes

The exchange process corresponds to the reactions

$$Hsp70\cdot\text{ATP} + \text{ADP} \underset{k^+_{ATP}}{\overset{k^-_{ATP}}{\rightleftarrows}} Hsp70 + \text{ATP} + \text{ADP} \underset{k^-_{ADP}}{\overset{k^+_{ADP}}{\rightleftarrows}} Hsp70\cdot\text{ADP} + \text{ATP}$$

The corresponding rate equations are

$$\frac{d[Hsp70\cdot\text{ATP}]}{dt} = -k^-_{ATP}[Hsp70\cdot\text{ATP}] + k^+_{ATP}[\text{ATP}][Hsp70]$$

$$\frac{d[Hsp70]}{dt} = -\left(k^+_{ATP}[\text{ATP}] + k^+_{ADP}[\text{ADP}]\right)[Hsp70] + k^-_{ATP}[Hsp70\cdot\text{ATP}]$$
$$+ k^-_{ADP}[Hsp70\cdot\text{ADP}]$$

$$\frac{d[Hsp70\cdot\text{ADP}]}{dt} = -k^-_{ADP}[Hsp70\cdot\text{ADP}] + k^+_{ADP}[\text{ADP}][Hsp70]$$

where $k^-_{ATP}$ and $k^-_{ADP}$ are the ATP and ADP release rates, respectively, and $k^+_{ATP}$ and $k^+_{ADP}$ are the ATP and ADP binding rates, respectively.

In steady state we can obtain the concentration of the nucleotide-free state from the second equation, and substitute it in the equations for the nucleotide-bound state concentrations, obtaining, for example (from the first equation)

$$\frac{d[Hsp70\cdot\text{ATP}]}{dt} = -k^-_{ATP}\frac{k^+_{ADP}[\text{ADP}]}{k^+_{ATP}[\text{ATP}] + k^+_{ADP}[\text{ADP}]}[Hsp70\cdot\text{ATP}]$$
$$+ k^-_{ADP}\frac{k^+_{ATP}[\text{ATP}]}{k^+_{ATP}[\text{ATP}] + k^+_{ADP}[\text{ADP}]}[Hsp70\cdot\text{ADP}]$$

and analogously for the equation for [Hsp70·ADP].

As a consequence, the expressions for the effective exchange rates $k^{ex}_{TD}$ and $k^{ex}_{DT}$ are

$$k^{ex}_{TD} = k^-_{ATP}\frac{k^+_{ADP}[\text{ADP}]}{k^+_{ATP}[\text{ATP}] + k^+_{ADP}[\text{ADP}]}$$

$$k_{DT}^{ex} = k_{ADP}^{-} \frac{k_{ATP}^{+}[\text{ATP}]}{k_{ATP}^{+}[\text{ATP}] + k_{ADP}^{+}[\text{ADP}]}$$

Typically the nucleotide release rates are much slower than the nucleotide binding rates (*Table 1*), and the above expressions correctly capture that nucleotide release sets upper bounds for the rates, which are then further modulated by the partitioning between ATP and ADP binding. Importantly, the equilibrium constant between the ATP-bound and ADP-bound states is correctly reproduced by the ratio $k_{TD}^{ex}/k_{DT}^{ex}$. Analogous expressions can be obtained for the exchange process in the presence of a bound substrate.

## Coarse graining of the co-chaperone action

The acceleration of hydrolysis by JDPs is described here as a modulation of $k_h^s$ in the range $k_h < k_h^s < k_{h,max}^S$, where $k_{h,max}^S$ is the maximal experimentally determined JDP/substrate accelerated hydrolysis rate. In this work we extend our analysis also to larger values of $k_h^s$ for completeness. A more complete description of the underlying co-chaperone binding/unbinding reactions would only overburden the present model without providing further insights. The dimerization of DnaJ into DnaJ$_2$ has also been considered as implicit.

## Parameters of the model

The rates that we have used to solve the model equations and for the data in *Figure 2,B* have been taken from studies of the DnaK/DnaJ/GrpE system, consistently with previous modeling (*Hu et al., 2006*), and are reported in *Table 1*. The value of $k_{TD}^{ex,S}$ has been obtained from the other parameters, using the relation

$$k_{TD}^{ex,S} = k_{TD}^{ex} \frac{K_D^{(\text{ATP})} k_{DT}^{ex,S}}{K_D^{(\text{ADP})} k_{DT}^{ex}}$$

which holds for a cycle at thermodynamic equilibrium because the free energy difference accumulated over a cycle is ΔG = 0 (*Ge et al., 2012*). The upper bound for the hydrolysis acceleration induced by JDPs ($k_{h,max}^S$) corresponds to the hydrolysis rate experimentally observed in saturation of DnaJ concomitant with the presence of the substrate (*Laufen et al., 1999*).

The rate of synthesis of ATP from ADP (the microscopic reverse of hydrolysis) is neglected here because it was experimentally proven to be below the level of detectability even in the presence of the hydrolysis-accelerating JDP cochaperones (*Russell et al., 1998*), *that is* $k_{synth} \leq 10^{-6}$ s$^{-1}$.

In all the calculations the chaperone concentration is 40 μM. The ratio [ATP]/[ADP] = 10 has been used throughout the calculations, approximately matching the physiological ratio.

## Equilibrium solution of the Hsp70 cycle

At equilibrium the *detailed balance* rule holds, which implies that the scheme in *Figure 1* can be solved by balancing each branch of the cycle individually. $K_{eff}^{eq}$ corresponds to the solution of the system of equations

$$k_{ATP}^{on}[\text{S}][\text{Hsp70·ATP}] = k_{ATP}^{off}[\text{Hsp70·ATP·S}]$$

$$k_{TD}^{ex}[\text{Hsp70·ATP}] = k_{DT}[\text{Hsp70·ADP}]$$

$$k_{ADP}^{on}[\text{S}][\text{Hsp70·ADP}] = k_{ADP}^{off}[\text{Hsp70·ADP·S}]$$

$$k_h^S[\text{Hsp70·ATP·S}] = k_{DT}^S[\text{Hsp70·ADP·S}]$$

$$[\text{S}] = [\text{S}]_{tot} - ([\text{Hsp70·ATP·S}] + [\text{Hsp70·ADP·S}])$$

where [S]$_{tot}$ is the total substrate concentration.

## Non-equilibrium solution of the Hsp70 cycle

In non-equilibrium the *detailed balance* rule does not hold anymore, and individual branches do not lead to a solution of the full cycle. Yet, the steady-state solution still exists and can be found solving the steady-state mass-balance equations for the scheme in *Figure 1*, namely

$$k_{ATP}^{on}[S][Hsp70 \cdot ATP] - \left(k_h^S + k_{ATP}^{off}\right)[Hsp70 \cdot ATP \cdot S] + k_{DT}^S[Hsp70 \cdot ADP \cdot S] = 0$$

$$k_{ATP}^{off,S}[Hsp70 \cdot ATP \cdot S] - \left(k_h + k_{ATP}^{on}[S]\right)[Hsp70 \cdot ATP] + k_{DT}[Hsp70 \cdot ADP] = 0$$

$$k_{ADP}^{on}[S][Hsp70 \cdot ADP] - \left(k_{DT}^S + k_{ADP}^{off}\right)[Hsp70 \cdot ADP \cdot S] + k_h^S[Hsp70 \cdot ATP \cdot S] = 0$$

$$k_{ADP}^{off,S}[Hsp70 \cdot ADP \cdot S] - \left(k_{DT} + k_{ADP}^{on}[S]\right)[Hsp70 \cdot ADP] + k_h[Hsp70 \cdot ATP] = 0$$

$$[S] = [S]_{tot} - \left([Hsp70 \cdot ATP \cdot S] + [Hsp70 \cdot ADP \cdot S]\right)$$

## Acknowledgements

AB thanks the Swiss National Science Foundation for financial support under the grant PZ00P2_136856. We thank SJ Landry for useful discussions.

## Additional information

### Funding

| Funder | Grant reference number | Author |
| --- | --- | --- |
| Swiss National Science Foundation | PZ00P2_136856 | Alessandro Barducci |

The funder had no role in study design, data collection and interpretation, or the decision to submit the work for publication.

### Author contributions

PDLR, Conception of the model Solution of the model, Drafting and revising the article; AB, Conception of the model, Solution of the model, Drafting and revising the article

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
