## [Decision Letter]

Thank you for sending your work entitled “Hsp70 chaperones as non-equilibrium machines: ultra-affinity from energy consumption” for consideration at *eLife*. Your article has been favorably evaluated by a Senior editor, a Guest Editor (Arup Chakraborty), and 2 peer reviewers.

The Guest Editor and the reviewers discussed their comments before we reached this decision, and the Guest Editor has assembled the following comments to help you prepare a revised submission.

Both reviewers and the Guest Editor found the paper to contain a number of interesting points, and it is an important topic being explored by others in the biophysics community. However, the reviewers raised some important points that need to be addressed in a revision. These do not require new analyses, but rather providing a clear articulation of some of the calculations, and justifying the assumptions and claims made in the paper. The reviews of the two reviewers are provided below in their entirety. Please address each of the itemized points in the resubmission, and briefly indicate how they have been addressed.

*Reviewer*
*#1:*

In their paper “Hsp70 chaperones as non-equilibrium machines: ultra-affinity from energy consumption”, De los Rios and Berducci address an interesting problem. What is the purpose of energy consumption in the Hsp70 protein system? The authors argue that energy consumption can increase the binding affinity of heat shock proteins for their substrates. The work is potentially interesting, and I think very clearly presented. However, in my opinion, it is not the rigorous theoretical study I would expect of a theoretical paper in a leading journal. In particular, it is unclear in what kinetic regimes their mechanism works in and how general it is beyond the parameters used to model the Hsp70 system.

Energy consumption has become an important topic of research in a variety of settings: for understanding adaptation in cells (see recent paper by Tu group in Nature Physics), sensing external concentrations (see work by Mehta and Schwab PNAS and ten Wolde group PRL 2012), and genetic switches (see paper by Qian group PRL 2005). More generally, the Qian and collaborators have developed a simple framework for understanding non-equilibrium thermodynamics for an open chemical system that is imported in this study.

The Hsp70 system is a special case of a four-state futile cycle analyzed in many works by Qian and has been generalized to stochastic settings where reactants occur is small numbers in many of the papers cited above. Thus, technically the work is straight forward, if simple, application of what are now somewhat standard techniques in this emerging field.

Nonetheless, I think the paper has a potentially interesting insight about non-equilibrium binding affinities. For this reason, I think it is potentially worth publishing. However, I have several misgivings about it in its present form that must be addressed. In particular, it is clear that their scheme relies on a separation of time scales for kinetics of the two states of the Hsp70 protein. However, it is not clear how general this mechanism is? What is the required separation of time scales? Instead of just plugging in parameters from the literature for Hsp70, it would be much more instructive if the authors made phase diagrams as a function of kinetic parameters and showed when there mechanism holds. The same comments hold for what relative concentrations of protein and substrates the mechanism holds for. I really would just like to have a clear idea when the mechanism they describe works. This is the minimum I would expect from a good theory paper.

Secondly, the authors rely on ’bulk’ thermodynamics that ignore effects of small protein numbers and substrates. This is probably justified in the Hsp70 system but nonetheless it would be nice to have a brief discussion about this point as it might be important for the applicability of the mechanism to other systems.

*Reviewer*
*#2:*

The authors show that when there are many different reaction pathways that accomplish the same net reaction, the effective dissociation constant of the net reaction, considering all pathways, can be less than the dissociation constant of any individual reaction pathway in isolation -provided there is energy dissipation. They then use this result, termed ’ultra-affinity,’ to understand an experimental observation on HSP chaperones. I found this paper very interesting to read.

I have two major concerns. Firstly, I have a concern about the novelty of their theoretical advance (apparent affinity can exceed individual affinities): is ultra-affinity related to kinetic proofreading? This concern does not touch the part of their paper that applies the result to chaperones.

Secondly, I have a concern about the correctness of their energy calculations: The authors inappropriately coarse-grained the energy? On this point, I think the calculations can be fixed, if indeed incorrect, without affecting the major conclusions of the manuscript.

Major comments:

1) Question concerning novelty: The authors show that “energy consumption can indeed decrease the dissociation constant ... with respect to an equilibrium scenario.” This phenomenon appears to have similarities to the well-known phenomenon of kinetic proofreading, which involves differential selectivity not explainable by equilibrium affinities. I would find it helpful if the authors would discuss their results in the context of kinetic proofreading, which does not seem to be mentioned in the current manuscript.

2) I have a question about whether gamma represents a dissipated energy as claimed. I understand the general argument that the dissipated energy across a cycle is equal to the ratio of the forward rates over the reverse rates across the cycle. However, I am only familiar with this argument when the forward and reverse rates are the rates for reactions that are the microscopic reverse of each other. When the rates represent coarse-grained rates, it is not clear to me that their ratio represents a free energy. In particular one of the ratios in [Disp-formula equ2] is k_{TD}/k_{DT} /approx (k_{TD}^{ex} + k_h)/(k_{DT} + k_h^r), which is not the ratio of microscopic rates and therefore does not seem like an energy. (Here I denote by k_h^r the – presumably small – rate of the reaction that microscopically reverses k_h). I would guess that there are different microscopic cycles in the problem, one that has 0 free energy drop (via the equilibrium reactions) and then one that has a free energy drop of G_ATP - G_ADP, and that the relevant quantity would be the (flux-) weighted average of the free-energy drops across these reactions. However, I have not been able to read the cited manuscript (Ge et al.) on account of its length, so I am not sure what the argument in that manuscript is. However, the absence of k_h^r in the model formulation may be concerning, as small rates can be neglected for calculating certain properties (e.g. the steady state levels of the species) but can have a major effect on the energy calculated from rates. (That is, making an already small rate 10 times smaller will not affect the usual observables in the system, but it will require more energy dissipation.)

3) The result of ultra-affinity seems fundamental – in support of the importance of the paper – but it relies on techniques that have been around for decades (Schnakenberg's paper, Hill's book on Free Energy), so it is tempting to wonder whether it is truly a novel result. In comment 1 above, I asked the authors to discuss kinetic proofreading in more detail. Here, I just want to comment that, in general, a more thorough discussion of the literature on the role of energy in biochemical systems may help to place the theoretical result more solidly as novel, if the authors feel they can do it without pulling focus from their discussion of the chaperone system.

4) I would find it helpful to have some additional detail about the calculations behind Figure 2. It wasn't clear to me which rate constants were varied to vary gamma in [Disp-formula equ2]. (I think it is k_h.) Also, I didn't see how the chaperone concentration entered [Disp-formula equ2] or 3, and so I didn't understand why the results in Figure 2 depended on the chaperone concentration. I apologize if I missed these details.

---

## [Author Response]

Reviewer #1:

*[…] I have several misgivings about it in its present form that must be addressed. In particular, it is clear that their scheme relies on a separation of time scales for kinetics of the two states of the Hsp70 protein. However, it is not clear how general this mechanism is? What is the required separation of time scales? Instead of just plugging in parameters from the literature for Hsp70, it would be much more instructive if the authors made phase diagrams as a function of kinetic parameters and showed when there mechanism holds. The same comments hold for what relative concentrations of protein and substrates the mechanism holds for. I really would just like to have a clear idea when the mechanism they describe works. This is the minimum I would expect from a good theory paper*.

We thank the reviewer for the suggestion. In the revised version, we analyzed more carefully both the effect of the separation of time scales and the effect of the relative concentrations of chaperones and substrates by means of phase diagrams (Figure 3). We believe that this more thorough exploration of the space of parameters, which is not limited to experimental Hsp70 data, addresses the questions raised by the reviewers. A completely exhaustive analysis of a four- state cycle would be beyond the scope of this work that focuses on the Hsp70 system and is aimed to a broad readership.

*Secondly, the authors rely on “bulk” thermodynamics that ignore effects of small protein numbers and substrates. This is probably justified in the Hsp70 system but nonetheless it would be nice to have a brief discussion about this point as it might be important for the applicability of the mechanism to other systems*.

The reviewer’s comment is correct and we added a brief discussion about the need of a stochastic description for less abundant proteins at the beginning of the Materials and method sections.

*Reviewer*
*#2:*

*1) The authors show that “energy consumption can indeed decrease the dissociation constant ... with respect to an equilibrium scenario.” This phenomenon appears to have (at least superficial) similarities to the well-known phenomenon of kinetic proofreading, which involves (differential) selectivity not explainable by equilibrium affinities. I would find it helpful if the authors would discuss their results in the context of kinetic proofreading, which does not seem to be mentioned in the current manuscript*.

We reply here to both point 1 and point 3 (see below) raised by referee 2. We thank the referee for the suggestions. In the revised manuscript we briefly discussed the general relevance of non-equilibrium thermodynamics in functional biochemical cycle and mentioned a few recent theoretical studies about this topic. Particularly, we sketched the major analogies and differences between the kinetic proofreading concept and the ultra-affinity proposed here.

*2) I have a question about whether gamma represents a dissipated energy as claimed. I understand the general argument that the dissipated energy across a cycle is equal to the ratio of the forward rates over the reverse rates across the cycle. However, I am only familiar with this argument when the forward and reverse rates are the rates for reactions that are the microscopic reverse of each other. When the rates represent coarse-grained rates, it is not clear to me that their ratio represents a free energy. In particular one of the ratios in*
[Disp-formula equ2]
*is k_{TD}/k_{DT} /approx (k_{TD}^{ex} + k_h)/(k_{DT} + k_h^r) which is not the ratio of microscopic rates and therefore does not seem like an energy. (Here I denote by k_h^r the - presumably small - rate of the reaction that microscopically reverses k_h). I would guess that there are different microscopic cycles in the problem, one that has 0 free energy drop (via the equilibrium reactions) and then one that has a free energy drop of G_ATP - G_ADP, and that the relevant quantity would be the (flux-) weighted average of the free-energy drops across these reactions. However, I have not been able to read the cited manuscript (Ge et al.) on account of its length, so I am not sure what the argument in that manuscript is. However, the absence of k_h^r in the model formulation may be concerning, as small rates can be neglected for calculating certain properties (e.g. the steady state levels of the species) but can have a major effect on the energy calculated from rates. (That is, making an already small rate 10 times smaller will not affect the usual observables in the system, but it will require more*
*energy dissipation.)*

We thank the reviewer for their insightful comment. Gamma as defined in our coarse-grained cycle does not straightforwardly correspond to dissipated energy. Therefore for the sake of clarity we used the ratio of the basal hydrolysis rate (kh) and the substrate accelerated one (khS) as key parameter to study the system behavior. Indeed, the latter is the quantity controlled at cellular level by means of substrate/cochaperone interactions. Moreover, the ratio khS/kh is directly related to energy consumption measured as ATP molecules hydrolyzed per second per chaperone (see Figure 2). Thus, we believe that the new presentation of our results in terms of experimentally measurable quantities makes our work accessible to a broader audience without affecting our main conclusion (*i.e.* enhanced affinity by means of energy-consumption). Regarding the reviewer’s concern about the possible role of ATP-synthesis rate, we point out that the latter process has been proved to be below the level of detectability implying a molecular rate below 10-6 s-1. We have checked that the consequences of a synthesis process with such a low rate in our scheme are negligible both for the steady-state concentrations and for energy consumption.

*3) [… In] general, a more thorough discussion of the literature on the role of energy in biochemical systems may help to place the theoretical result more solidly as novel, if the authors feel they can do it without pulling focus from their discussion of the chaperone system*.

See reply to point 1.

*4) I would find it helpful to have some additional detail about the calculations behind*
Figure 2*. It wasn't clear to me which rate constants were varied to vary gamma in*
[Disp-formula equ2]*. (I think it is k_h.) Also, I didn't see how the chaperone concentration entered*
[Disp-formula equ2]
*or 3, and so I didn't understand why the results in*
Figure 2
*depended on the chaperone concentration. I apologize if I missed these details*.

As the reviewer correctly pointed out, we varied gamma by changing *khS.* In the new version of the manuscript, the central role of *khS* is obviously even more explicit. The total chaperone concentration enters the calculations by means of its effects on the five mass-balance non-linear coupled equations that solve the system (see Materials and methods). Technically, the first four equations are a linear system that can be solved as a function of the free substrate concentration. An iterative scheme ensured that the latter was consistent with the fifth equation.